# RMR-Related *DNAJC6* Expression Suppresses Adipogenesis in 3T3-L1 Cells

**DOI:** 10.3390/cells11081331

**Published:** 2022-04-13

**Authors:** Juhee Kim, Myoungsook Lee

**Affiliations:** 1Department of Food & Nutrition, Sungshin Women’s University, Seoul 01133, Korea; juheeq.kim@samsung.com; 2Medical Research Institute, Kangbuk Samsung Hospital, Seoul 04514, Korea; 3Research Institute of Obesity Sciences, Sungshin Women’s University, Kangbuk-ku, Seoul 01133, Korea

**Keywords:** obesity, *DNAJC6*, insulin pathway, inflammatory cytokines, autophagy, thermogenesis

## Abstract

Obesity causes various complications such as type 2 diabetes, hypertension, fatty liver, cardiovascular diseases, and cancer. In a pilot GWAS study, we screened the *DNAJC6* gene which is significantly related to the resting metabolic rate (RMR) in childhood obesity. With *DNAJC6*-overexpressed 3T3-L1 cells (Tg*^Hsp^*), we investigated the new obesity mechanism caused by an energy imbalance. After differentiation, lipid droplets (Oil red O staining) were not formed in Tg*^Hsp^* cells compared to the control. Tg*^Hsp^* preadipocyte fibroblast morphology was also not clearly observed in the cell morphology assay (DAPI/BODIPY). In Tg*^Hsp^* cells, the expression of PPARγ, C/EBPα, and aP2 (adipogenesis-related biomarkers) decreased 3-, 39-, and 200-fold, respectively. The expression of the adipokines leptin and adiponectin from adipose tissues also decreased 2.4- and 840-fold, respectively. In addition, the levels of pHSL(Ser563) and free glycerol, which are involved in lipolysis, were significantly lower in Tg*^Hsp^* cells than in the control. The reduction in insulin receptor expression in Tg*^Hsp^* cells suppressed insulin signaling systems such as AKT phosphorylation, and GLUT4 expression. Degradation of IRS-1 in 3T3-L1 adipocytes was caused by chronic exposure to insulin, but not Tg*^Hsp^*. Mitochondrial functions such as oxygen consumption and ATP production, as well as proton leak and UCP1 protein expression, decreased in Tg*^Hsp^* cells compared to the control. Moreover, autophagy was observed by increasing autophagosomal proteins, LC3, on Day 8 of differentiation in Tg*^Hsp^* cells. Through our first report on the *DNAJC6* gene related to RMR*,* we found a new mechanism related to energy metabolism in obesity. *DNAJC6* expression positively suppressed adipogenesis, leading to the subsequent resistance of lipolysis, adipokine expression, insulin signaling, and mitochondrial functions.

## 1. Introduction

Obesity, a growing concern worldwide, is a complex health disease affected by various environmental factors such as heredity factors, lifestyle, and dietary habits. Studies are being actively conducted to identify obesity-related environmental factors [1,2]. Following the Human Genome Project, genome-wide association studies (GWASs) are being performed to investigate genes related to obesity mechanisms using new technologies [3,4].

Our pilot GWAS reported that two genes, mitogen-activated protein kinase, kinase 6 (*MAP2K6*) and DnaJ Heat shock Protein Family Member C6 (*DNAJC6)*, were distinguishingly related to both the resting metabolic rate (RMR) and body mass index (BMI) as biomarkers of obesity in children. The results of a human study on *MAP2K6* gene variation and the roles of *MAP2K6* overexpression in cell and animal studies were published [5,6,7]. *DNAJC6* encodes auxilin, a neuronal protein that functions specifically in the pathway of clathrin-mediated endocytosis [8]. *DNAJC6* belongs to the evolutionarily conserved *DNAJ/HSP40* family of proteins, which regulate molecular chaperone activity by stimulating ATPase activity [9]. The *DNAJC6* gene is present in organisms and cells and is stimulated by various environmental stresses such as heat, cold, and ultraviolet rays. Several studies reported that *DNAJC6* mutation might be related to the risk factors for the early onset of degenerative diseases, such as Parkinson’s disease and Alzheimer’s disease [10,11,12]. However, few studies have shown an association between *DNAJC6* mutation and obesity. Vauthier et al. found that 7-year-old children with a homozygous 80 kb deletion in the chromosomal 1p31.3 region exhibited early-onset obesity, mental retardation, and epilepsy [13]. In an animal study, *Dnajc6*-null mice had a high rate of early postnatal mortality, although surviving pups had a normal life span despite a decrease in body weight [14]. However, there has been no mechanistic study indicating that *DNAJC6* deficiency or mutation leads to obesity thus far. Although we found that *DNAJC6* is a gene related to RMR, it is necessary to have mechanistic research or systematic reviews on lipid synthesis and degradation for energy metabolism, inflammation, insulin signaling, etc.

The purpose of this study was to investigate new obesity mechanisms related to energy metabolism such as lipid metabolism (adipogenesis and lipolysis), adipokine expression, insulin signaling, and mitochondrial function, using *DNAJC6*-overexpressed 3T3-L1 preadipocytes. This study is the first report to show the effects of the RMR-related *DNAJC6* gene on obesogenic environments, and it will provide the basic data for obesity research on customized prevention and treatment.

## 2. Materials and Methods

### 2.1. Cell Culture

3T3-L1 preadipocytes were purchased from the American Type Culture Collection (ATCC, Manassas, VA, USA) and cultured in Dulbecco’s Modified Eagle Medium (DMEM, Welgene, Gyeongsan, Korea) containing 1% penicillin/streptomycin (P/S, Welgene) and 10% Bovine Calf Serum (BCS, Thermo Fisher, Waltham, MA, USA) under 5% CO₂ at 37 °C. To induce differentiation (Day 0), 1 μM dexamethasone, 10 μg/mL insulin, and 0.5 mM 3-isobutyl-1-methylxanthine (MDI, Cat #D1756, #I9278, #I7018, Sigma-Aldrich, St. Louis, MO, USA) were added to the culture medium of high-glucose DMEM containing 1% P/S and 10% Fetal Bovine Serum (FBS, Capricorn Scientific GmbH, Ebsdorfergrund, Germany). After two days of differentiation (Day 2), the cells were cultured in 10% FBS high-glucose DMEM containing 10 μg/mL insulin. After four days of differentiation (Day 4), the 10% FBS high-glucose DMEM was changed every 48 h. The cells were harvested after eight days of differentiation (Day 8), and *DNAJC6*-overexpressed cells (Tg*^Hsp^*) and 3T3-L1 cells (control group) were compared. An MTT assay (Duchefa, Haarlem, The Netherlands) was conducted to evaluate cell viability. Absorbance was measured at 570–630 nm using a microplate reader.

### 2.2. DNAJC6 Transfection

3T3-L1 preadipocytes were transfected with *DNAJC6* using a chemical method (L3000015, Thermo Fisher, Waltham, MA, USA). To increase the transfection efficiency, serum-free DMEM was used. A DNA master mix was prepared by mixing Opti-MEM, 2500 ng *DNAJC6* DNA (OriGene, Rockville, MD, USA), P3000 reagent, and lipofectamine 3000 reagent. After reacting at room temperature for 15 min, the DNA master mix was dispensed into cells and cultured for 3–4 h. To remove untransfected cells during the transfection process, cells was cultured in DMEM containing 500 μg/mL of G418 (Cat #10131035, Thermo Fisher, Waltham, MA, USA) for 20 h under the same conditions described above. Thereafter, Tg*^Hsp^* cells were cultured using the same process and conditions of differentiation as those of the control group (Days 0–8).

### 2.3. RT-PCR to Confirm Transfection

TRIzol reagent (Cat #15596018, Thermo Fisher, Waltham, MA, USA) was used to collect cells, and an RNA pellet was centrifuged. The pellet was completely dissolved in RNase-free water and quantified using a Nano-Drop spectrophotometer. RNA (1 μg), dNTPs (Cat #4030, Takara Bio, Shiga, Japan), and a random primer (Invitrogen, San Diego, CA, USA) were mixed and heated at 65 °C for 5 min. A master mix was prepared with Superscript III Reverse Transcriptase, 5× First Strand buffer, and 0.1 M DTT (Cat #18080093, Invitrogen, Carlsbad, CA, USA), and the RNA mix and master mix were combined to prepare cDNA. PCR PreMix (Cat #K2036, Bioneer, Daejeon, Korea) was mixed with 19 µL of primer (Zenotech, Daejeon, Korea) and 1 µL of cDNA. The *DNAJC6* primer (forward: GTG TAC GGT GGG AGG TCT AT; reverse: CCG CCT TTC ACC ATG TCA AA) was heated at 94 °C for 4 min, followed by 30 cycles of 30 s at 94 °C, 30 s at 61 °C, and 40 s at 72 °C. A total of 20 µL of the prepared sample was loaded on an agarose gel and electrophoresed at 100 V for 30 min. The gel was analyzed using a Chemi Doc Imaging System (Bio-Rad Laboratories, Hercules, CA, USA).

### 2.4. Cell Staining

The accumulation of lipid droplets was evaluated using Oil red O (ORO). On Days 0 and 8 of differentiation, the medium and PBS were removed. The cells were fixed with 1 mL of 4% paraformaldehyde for 1 h at room temperature. The cells were stained with 1 mL of ORO solution for 1 h and then washed with PBS. The lipid droplets were observed using an optical microscope and extracted with lipid droplets assay dye extraction solution (Cat #10008984, Cayman Chemical Company, Ann Arbor, MI, USA). Absorbance was measured at 492 nm. The cells were stained with DAPI/BODIPY to confirm nuclear damage and adipogenesis following transfection. The cells were fixed with 1 mL of 4% paraformaldehyde and washed with PBS. For the staining of lipid droplets and nuclear damage, the cells were stained with BODIPY and DAPI for 30 min. After staining the cell nucleus, a wet mount was made with fluorescent mounting medium (DAKO, Glostrup, Denmark), and the fat contents and nuclei of cells were observed using a confocal laser microscope (Carl Zeiss, Oberkochen, Germany).

### 2.5. Adipogenesis and Lipolysis Evaluation

Adipogenesis in adipocytes of 3T3-L1 and Tg*^Hsp^* cells was confirmed by ORO staining, and the expression of adipogenesis-related genes was measured by Western blotting analysis. Free glycerol in the medium was measured to confirm the breakdown of accumulated fat. Samples were collected on Days 0 and 8 of differentiation and stored at 37 °C for use on the same day. A total of 240 µL of free glycerol reagent (Cat #F6428, Sigma-Aldrich, St. Louis, MO, USA) was pipetted into a 1.5 mL amber tube, and 30 µL each of standard, distilled water, and sample was added. The tube was inverted immediately and incubated at 37 °C for 5 min. A total of 100 µL of each sample was added to a 96-well plate and measured at 540 nm. The measured value of the sample was divided by the measured value of the standard solution and multiplied by 0.26 mg/mL to calculate the amount of free glycerol.

### 2.6. Preparation of Protein Extracts and Western Blot Analysis

The protein of cells was extracted using RIPA buffer (EBA-1149, Elpis Biotech, Daejeon, Korea) containing protease and phosphatase inhibitors. Protein samples were quantified through the Bradford assay. The samples were then separated by electrophoresis using 8–12% SDS and transferred to a PVDF membrane (Cat #10600021, GE Healthcare, Chicago, IL, USA). The protein samples were blocked with 3% bovine serum albumin (BSA, Cat #BSA100, Bovogen, Melbourne, Australia) and incubated at 4 °C for 16 h with the following primary antibodies: β-actin (E12-041, Enogene, Jiangsu, China), Peroxisome proliferator-activated receptor gamma (PPARγ, Cat #SC-7273, Santa Cruz, Dallas, TX, USA), CCAAT/enhancer-binding protein alpha (C/EBPα, Cat #SC-9314, Santa Cruz), Adipocyte Fatty acid-Binding Protein (aP2, Cat #SC-271529, Santa Cruz, Dallas, TX, USA), phospho-Hormone-sensitive Lipase (pHSL, Cat #4139S, CST, Beverly, MA, USA), HSL (Cat #4170S, CST, Beverly, MA, USA), insulin receptor-β (Cat #3025, CST, Beverly, MA, USA), Insulin receptor substrate (IRS-1, Cat #2382S, CST, Beverly, MA, USA), p-IRS-1 (Cat #2381S, CST, Beverly, MA, USA), Protein Kinase B (AKT, Cat #9272, CST, Beverly, MA, USA), p-AKT (Cat #9275, CST, Beverly, MA, USA), Glucose transporter type 4 (GLUT4, Cat #2213, CST, Beverly, MA, USA), leptin (Cat #675002, Bio Legend, San Diego, CA, USA), leptin receptor (LEPR, Cat #ab5593, Abcam, Cambridge, UK), adiponectin (Cat #2789S, CST, Beverly, MA, USA), Uncoupling Protein1 (UCP1, Cat #14670S, CST, Beverly, MA, USA), and Microtubule-associated protein 1A/1B-light chain 3 (LC3-I/LC3-II, Cat #12741S, CST, Beverly, MA, USA). Subsequently, secondary antibodies were incubated at room temperature. Analysis was conducted using a ChemiDoc imaging system and Image Lab software (Bio-Rad Laboratories, Hercules, CA, USA).

### 2.7. Oxygen Consumption Test (Mitochondrial Function Test)

A Seahorse XF Cell Mito Stress Test Kit (Cat #Kit103015-100, Agilent Technologies, CA, USA) was used to analyze mitochondrial activity and oxygen consumption. 3T3-L1 and Tg*^Hsp^* cells in an XF Cell Culture Microplate were cultured under the same conditions and procedures as those for adipocyte differentiation. The calibrant solution was preheated for hydration 24 h before evaluating mito stress. Mitochondrial respiration was measured on Day 8 using a seahorse XFe24 analyzer (Agilent Technologies, San Jose, CA, USA). Oligomycin, carbonyl cyanide-*p*-trifluoromethoxyphenylhydrazone (FCCP), and rotenone/antimycin A (Rot/AA), which were injected into the mitochondria, were dissolved in XF media and injected into the drug port. After calibrating the calibrator of the Cell Mito Stress measuring instrument, oxygen consumption was measured by placing the drop port and microplate with cultured cells. Oligomycin, FCCP, and Rot/AA were injected, and each section was measured for 5 min. Mitochondrial function was calculated for each section. After measurement, cellular proteins were extracted, quantified, and normalized.

### 2.8. Statistical Analysis

Collected data were analyzed using Statistical Package for the Social Sciences software version 25.0 (SPSS Inc., Chicago, IL, USA). Student’s *t*-test or one-way analysis of variance (ANOVA) with Duncan’s multiple range test was performed to analyze significant differences between groups. The results were expressed as means ± standard deviations (SD). A value of *p* < 0.05 was considered statistically significant for differences between the means, which were described as a, b, c, and d, which indicate significant differences between the groups. Graphs were plotted using GraphPad Prism (version 8.0.1, GraphPad Software, La Jolla, CA, USA).

## 3. Results

### 3.1. DNAJC6 Transfection Confirmation and Cell Proliferation

To confirm transfection, RT-PCR was performed on Day 0 of differentiation. A higher mRNA expression was observed in Tg*^Hsp^* cells than in 3T3-L1 cells (Figure 1A). DNAJC6 expression in 3T3-L1 cells was 0.99 ± 0.24 on Day 0 of differentiation (3T3-0) and 0.54 ± 0.17 on Day 8 of differentiation (3T3-8), suggesting an approximate 1.8-fold decrease in *DNAJC6* levels due to differentiation. In contrast, DNAJC6 protein expression remained the same in Tg*^Hsp^* cells after differentiation. An MTT assay was conducted to assess the appropriate concentration of probiotics to increase *DNAJC6* transfection efficiency. Tg*^Hsp^* cells showed 59% cell viability after treatment with 700 μg/mL of G418; this value was higher than that of the control group (Figure 1B). Based on the cell viability test results, 500 μg/mL of G418 was used for *DNAJC6* transfection. The cells were stained with DAPI/BODIPY to confirm nuclear damage during *DNAJC6* transfection and differentiation. In 3T3-0 cells, the cytoplasm maintained the form of fibroblasts, and round lipids were formed around the cell nucleus after differentiation. On the other hand, the morphological characteristics of fibroblasts were not observed in Tg*^Hsp^*-0 cells, and lipids were not formed in Tg*^Hsp^*-8 cells (Figure 1C).

### 3.2. Adipogenesis and Lipolysis

To determine the effects of *DNAJC6* overexpression on lipid formation, cells were stained with ORO solution. In the control group, lipid levels increased 7-fold at the end of differentiation (Day 8) compared to those before differentiation. In Tg*^Hsp^* cells, no lipids were observed after differentiation. There was no significant difference between Tg*^Hsp^*-8 cells and Tg*^Hsp^*-0 cells with respect to the results obtained upon extracting and measuring stained lipids (Figure 2A). To confirm that the formation of lipids was inhibited in Tg*^Hsp^* cells, the expression of adipogenesis-related proteins was evaluated. The levels of the PPARγ, C/EBPα, and aP2 proteins were significantly higher in 3T3-8 cells than in 3T3-0 cells; however, there was no significant difference in Tg*^Hsp^* cells according to differentiation (Figure 2B).

To assess the degradation of intracellular triglycerides, HSL activity in cell lysates and free glycerol in growth media were measured. In 3T3−L1 cells, the expression of HSL and pHSL (Ser563) increased in a time-dependent manner. On the other hand, in Tg*^Hsp^* cells, there was no significant change. The level of pHSL (Ser563), which is an activated HSL, increased 1.73-fold in 3T3-8 cells without insulin compared to that in 3T3-4 cells; however, there was no significant change in Tg*^Hsp^* cells. This suggests that lipolysis did not occur in Tg*^Hsp^* cells (Figure 2C). Free glycerol secretion into the medium was significantly higher on Day 8 of differentiation than on Day 0 of differentiation in both 3T3-L1 and Tg*^Hsp^* cells. The amount of free glycerol in the medium increased 20-fold in 3T3-8 cells compared to that in 3T3-0 cells and 3-fold in Tg*^Hsp^*-8 cells compared to that in Tg*^Hsp^*-0 cells. However, compared to 3T3-8 cells, Tg*^Hsp^*-8 cells showed reduced expression of HSL and pHSL (Ser563) and a lower free glycerol content, suggesting that lipolysis was reduced.

### 3.3. Adipokine Production and Insulin Signaling

Leptin expression increased 2.5-fold in 3T3-8 cells compared to that in 3T3-0 cells. However, there was no significant change in leptin expression according to differentiation in Tg*^Hsp^* cells. Leptin expression in Tg*^Hsp^*-8 cells was 2.4-fold lower than that in 3T3-8 cells. LEPR expression showed a pattern similar to that of leptin expression and was approximately 4-fold higher in 3T3-8 cells than in Tg*^Hsp^*-8 cells. Adiponectin expression was low in 3T3-0, Tg*^Hsp^*-0, and Tg*^Hsp^*-8 cells and was significantly higher in 3T3-8 cells than in Tg*^Hsp^*-8 cells, by approximately 840-fold (Figure 3A).

Insulin receptor-β expression for glucose influx into adipocytes was significantly higher in 3T3-8 cells than in 3T3-0 cells. However, insulin receptor-β expression in Tg*^Hsp^*-8 cells was 0.969 ± 0.475, which was not significantly different from that in Tg*^Hsp^*-0 cells. In the control cells, the ratio of p-IRS (Ser307) vs. IRS-1 levels tended to be higher in 3T3-8 cells than in 3T3-0 cells; however, there was no significant change in IRS-1 activity in Tg*^Hsp^* cells with insulin stimulation. Threonine phosphorylation of AKT was higher in 3T3-8 cells than in 3T3-0 cells, but AKT phosphorylation as well as IRS-1 phosphorylation according to differentiation was not observed in Tg*^Hsp^* cells. GLUT4 levels were significantly higher in 3T3-8 cells than in 3T3-0 cells. However, GLUT4 expression in Tg*^Hsp^*-8 cells was 0.958 ± 0.624, which was lower compared to the 1.127 ± 0.301 in Tg*^Hsp^*-0 cells. GLUT4 protein expression in 3T3-8 cells was higher than that in 3T3-0, Tg*^Hsp^*-0, and Tg*^Hsp^*-8 cells by more than 10-fold (Figure 3B).

### 3.4. Mitochondrial Function (Thermogenesis) and Autophagy

We observed that insulin signaling was inhibited in Tg*^Hsp^* cells. Hence, oxygen consumption was measured. In the control group, the maximal respiratory capacity (216 pmol/min) was higher, by 3.6-fold, than the basal respiratory capacity (59 pmol/min). In contrast, in Tg*^Hsp^* cells, the maximal respiratory capacity (38 pmol/min) was 1.7-fold greater than the basal respiratory capacity (22 pmol/min). In the control group, the spare respiratory capacity was 157 pmol/min, which was 9.5-fold higher than the 16 pmol/min in Tg*^Hsp^* cells (Figure 4A). In Tg*^Hsp^* cells, proton leak decreased approximately 5-fold, and UCP1 expression was significantly lower than that in the control group (Figure 4B). On the other hand, coupling efficiency was significantly higher in Tg*^Hsp^* cells than in the control group. The LC3-II/LC3-I protein expression ratio was significantly higher in Tg*^Hsp^* cells than in 3T3-8 cells (Figure 4C). These findings suggest that Tg*^Hsp^* cells are protected by the uptake of nutrients through programmed cell death in the condition of nutrient deficiency.

## 4. Discussion

Heat shock proteins are present in all organisms and cells and are induced by various environmental stresses [15]. Among them, *DNAJC6* is a major risk factor for the early onset of Alzheimer’s and Parkinson’s diseases. From a previous GWAS, we found that two genes, *MAP2K6 (MEK6)* and *DNAJC6,* were related to childhood obesity due to energy metabolism imbalance, particularly RMR. Based on human studies, energy metabolism and inflammatory mechanisms related to body fat accumulation were investigated in cells and animals transfected with those genes. *MEK6* gene overexpression significantly induced fat accumulation with consistent results of increasing adipogenesis and decreasing adipolysis both in vitro and in vivo [6,7]. In previous studies, the mechanism of obesity induction mediated by *DNAJC6* was not established [16,17]. However, we found that the overexpression of *DNAJC6* in 3T3-L1 cells positively suppressed adipogenesis-related gene expression as well as inhibiting insulin signals and adipokine expression.

Compared to after the differentiation of 3T3-L1 cells, lipid formation was suppressed in the *DNAJC6*-overexpressed cells, and no morphological changes in the cells were observed. Adipogenesis is induced by the transcription of C/EBPβ, C/EBPδ, and PPARγ, and adipokine genes are expressed in the late stage of adipogenesis [18,19]. Pro-inflammatory adipokines, such as TNF-α and IL-6, promote inflammatory M1 adipocytes, but anti-inflammatory adipokines, such as adiponectin, induce anti-inflammatory M2 adipocytes. The interactions between these pro- and anti-inflammatory adipocytes may change lipid metabolism and insulin/leptin resistance in adipose tissues [20]. For the adipogenesis process in adipocytes, glucose influx is required through the GLUT4 transporter on the membranes with the activation of insulin cascade signals, IRS-1^ser307^-AKT^thr308^-GLUT4. [21,22]. Insulin/IGF-1 and TNF-α stimulated the phosphorylation of IRS-1 at Ser^307^ in 3T3-L1 preadipocytes and adipocytes. However, the distinct kinase pathways might converge at Ser^307^ to mediate the feedback inhibition of IRS-1 signaling to counter-regulate the insulin response in some conditions such as overexposure of insulin and dexamethasone, or oxidative stress [23]. In the *DNAJC6*-overexpressing group, the expression of both adipogenesis genes and adipokines was not changed while the insulin was treated. Defects in the insulin signaling system were more affected by the up/downregulation of the insulin receptor than the binding force between insulin and the receptor [24]. The reduction in the insulin/receptor complex due to *DNAJC6* overexpression subsequently suppressed the recruitments of the GLUT4 transporters from the intracellular stores. In the *DNAJC6*-overexpressing group, the inhibition of the insulin signaling system also inhibited leptin and LEPR expression, which increases in a dose-dependent manner and with the insulin treatment time [25]. The deleted region comprised the proximal promoter and exons 1 and 2 of the *LEPR* gene and exons 5 to 19 of the *DNAJC6* gene. This 80 kb deletion is consistent with previous observations that heterozygous human carriers of *leptin* or *LEPR* mutations are predisposed to overweight and obesity [13]. The regulation of leptin secretion homeostasis in proportion to the size and number of adipocytes is dependent on LEPRs (up/downregulation) that are present in most cells. Therefore, the mechanisms involving the endocrine, immune, cardiovascular, and respiratory systems, in addition to diet and body fat control, are broad and complex [26,27]. Similar to the insulin receptor, various studies have shown that the leptin receptor exhibits substrate resistance and a cross-talk mechanism acting on both insulin and leptin [28]. As adiponectin is regulated by PPARγ [29], adiponectin may not have been expressed in the *DNAJC6*-overexpressing group, in which the expression of adipogenesis-related proteins was suppressed.

Furthermore, in the *DNAJC6*-overexpressing group, the expression of pHSL (Ser563) involved in lipolysis, the free glycerol content, mitochondrial oxygen and energy consumption, and UCP1 protein expression were suppressed. HSL is activated by cyclin AMP-dependent protein kinases, whose expression increases during 3T3-L1 cell differentiation. HSL plays an important role in lipolysis [30,31]. However, in the *DNAJC6*-overexpressing group, lipolysis was inhibited, and lipids were not formed. Therefore, lipolysis and adipokine expression were not observed. The low spare respiratory capacity and small increase in the maximum respiratory capacity in the *DNAJC6*-overexpressing group suggest that mitochondria were already performing oxygen respiration at maximum capacity [32]. Proton leak induced by inhibition of ATP synthesis is an indicator of a potential thermogenesis mechanism [33,34] and is related to the basal metabolic rate (BMR). In the *DNAJC6*-overexpressing group, both proton leak and UCP1 expression decreased. These findings indicate that oxygen consumption was reduced in the *DNAJC6*-overexpressing group due to a decrease in the BMR [35]. Compared to the control group, the *DNAJC6*-overexpressing group showed a significantly higher coupling efficiency. This suggests that, although coupling respiration is more active than uncoupling respiration in the *DNAJC6*-overexpressing group, energy production is lower than that in the control group [34].

Since non-adipogenesis, non-glucose influx, non-ATP generation, and non-thermogenesis were found in the differentiation of *DNAJC6*-overexpressed cells, we need to know the next survival step in the case of continuous malnutrition, such as autophagy progress. The conversion rate from LC3-I to LC3-II as a biomarker of autophagosome formation was increased in the *DNAJC6*-overexpressed cells at the status of differentiation compared to the control. Mizushima reported that a simple comparison of LC3-I and LC3-II, or a summation of LC3-I and LC3-II for ratio determinations, may not be appropriate, and instead, the amount of LC3-II can be compared between samples [36]. Although they recommended LC3-II levels, which are clearly correlated with the numbers of autophagosomes, we used the ratio of LC3-II/LC-I because of the importance of the conversion rate of LC3-II from LC3-I. Without the metabolic rate for energy production in *DNAJC6*-overexpressed cells, autophagy/programmed cell death might have occurred to protect the cells from apoptosis [37]. We concluded that the lack of adipogenesis in the *DNAJC6*-overexpressing group might be caused by the inhibition of the following: expression of adipogenesis-related genes, glucose inflow due to defects in the insulin signaling system, and mitochondrial dysfunction.

Our limitation is that we did not confirm whether a null *DNAJC6* case using knock-out (KO) transfection or siRNA induces adipogenesis compared to *DNAJC6* overexpression. Therefore, we need animal experiments that use both cases of *DNAJC6* overexpression and KO groups to confirm the mechanism of lipid energy metabolism.

## 5. Conclusions

To date, there is a lack of studies on the mechanism of energy imbalance in obesity caused by overexpression or mutation of the *DNAJC6* gene, an RMR-related gene. Using *DNAJC6-*overexpressed 3T3-L1 preadipocytes, we observed that adipogenesis was inhibited, leading to subsequent suppression of lipolysis, adipokine expression, insulin signaling, and energy metabolism. In future studies, in vivo experiments using *DNAJC6* transgenic mice, based on these findings, may be conducted to identify the energy mechanisms to prevent obesity and provide basic data for research on personalized obesity prevention and treatment.

## Figures and Tables

**Figure 1 cells-11-01331-f001:**
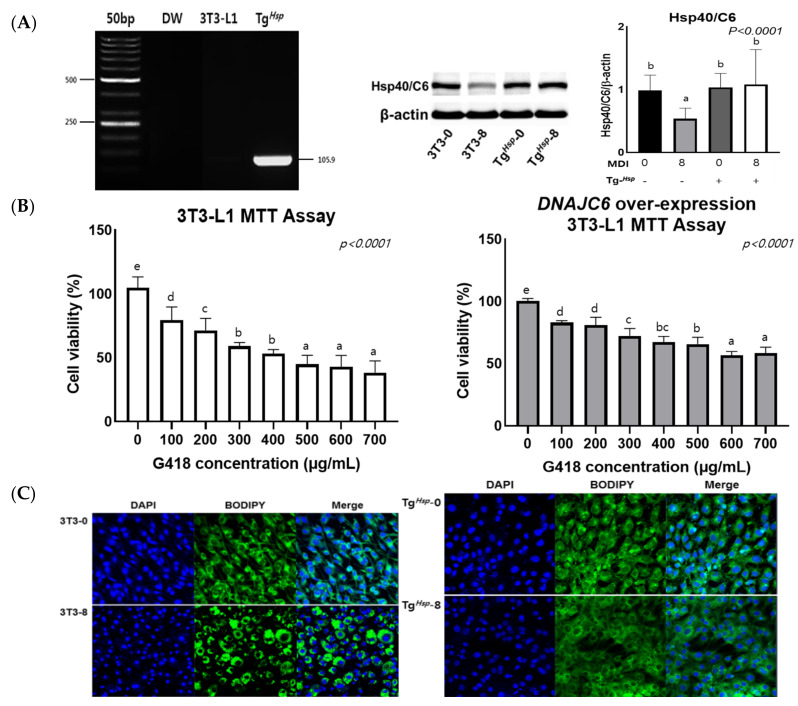
mRNA and protein expression of *DNAJC6* (**A**), and cell viability (**B**) in Tg*^Hsp^* cells compared to that in 3T3-L1 cells without nucleus impairment (**C**). Data were assessed by one-way ANOVA with Duncan’s test. Superscript letters (a, b, c, and d) indicate significant differences between groups at *p* < 0.05.

**Figure 2 cells-11-01331-f002:**
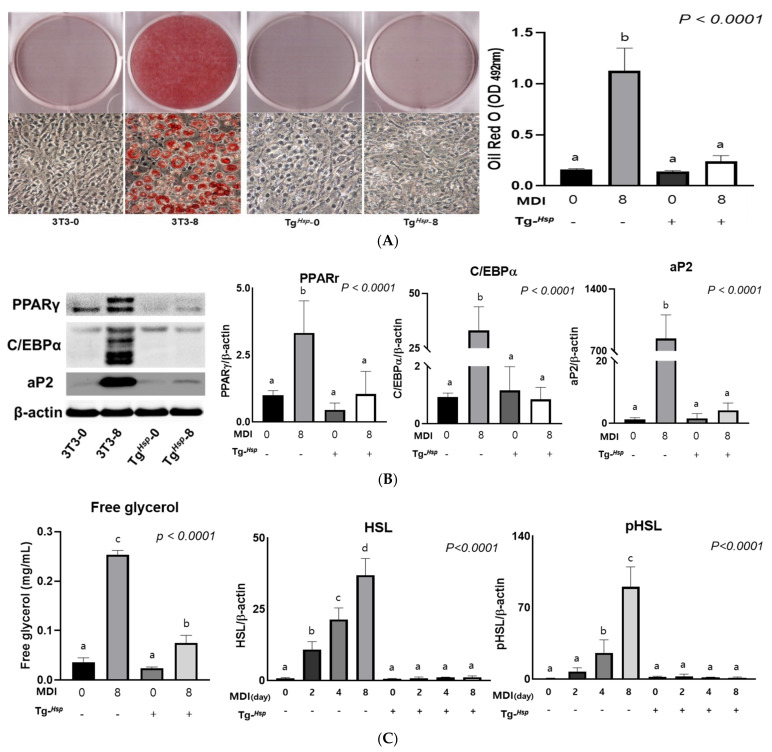
Effect of Tg*^Hsp^* in 3T3−L1 cells on lipid accumulation and lipolysis. (**A**) ORO staining on Days 0 and 8 of differentiation in 3T3-L1 and Tg*^Hsp^* cells. Absorbance of extracted ORO solution was measured at 492 nm. (**B**) Expression of adipogenesis biomarkers in 3T3-L1 and Tg*^Hsp^* cells. (**C**) Cell culture media were assayed for free glycerol. HSL activation was assessed on Days 0 and 8 of differentiation. The data are presented as means ± SDs of at least three replicates. Means with different superscripts are significantly different at *p* < 0.05 by Duncan’s multiple range test.

**Figure 3 cells-11-01331-f003:**
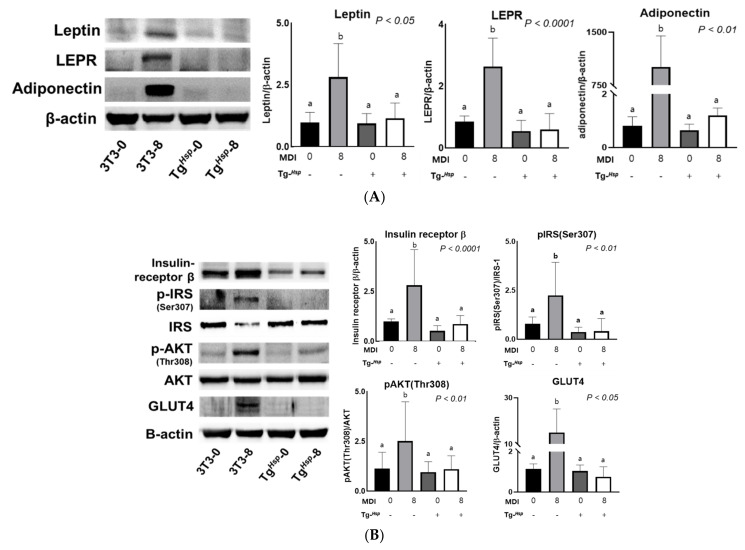
Expression of leptin, LEPR, and adiponectin in 3T3−L1 and Tg*^Hsp^* cells (**A**). Tg*^Hsp^* inhibits insulin signaling in 3T3−L1 cells (**B**). Protein levels of adipokines and insulin signals were analyzed by Western blotting. Data are presented as means ± SDs of at least three replicates. Means with different superscripts are significantly different at *p* < 0.05 by Duncan’s multiple range test.

**Figure 4 cells-11-01331-f004:**
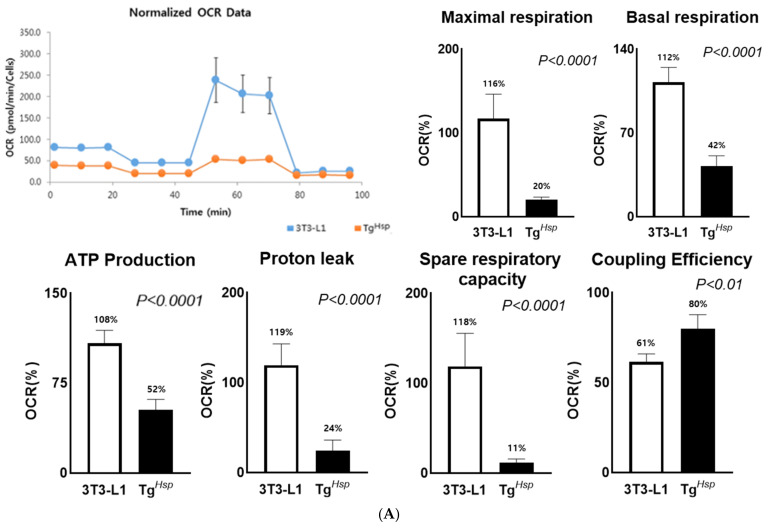
OCR in 3T3−L1 and Tg*^Hsp^* cells (**A**), and expression of UCP1 in 3T3−L1 and Tg*^Hsp^* cells (**B**); Tg*^Hsp^* induced the expression of autophagosomal proteins in 3T3−L1 and Tg*^Hsp^* cells (**C**). To compare the OCR between 3T3−L1 and Tg*^Hsp^*, these two types of cells were measured on Day 8 of differentiation. OCR measurement was confirmed using the Seahorse XF Mito Stress Test Kit. OCR was measured using 1.0 × 10^4^ 3T3−L1 cells and 2.0 × 10^4^ Tg*^Hsp^* cells. Basal respiratory capacity, maximal respiratory capacity, spare respiratory capacity, proton leak, and ATP production were calculated using OCR and are shown as graphs. Data are presented as means ± SDs of five replicates. Means with different superscripts are significantly different at *p* < 0.05 by a *t*-test. (**A**) Protein levels for UCP1 (**B**) and LC3−I and LC3−II (**C**) were analyzed by Western blotting. The LC3−II/LC3−I ratio is shown on the right of graph **C**. Data are presented as means ± SDs of at least three replicates. Means with different superscripts are significantly different at *p* < 0.05 by Duncan’s multiple range test.

## Data Availability

Not applicable.

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
