# Peer review of "RMR-Related DNAJC6 Expression Suppresses Adipogenesis in 3T3-L1 Cells"

_cells, 2022, doi:10.3390/cells11081331_

Round 1

Reviewer 1 Report

The mechanisms of energy metabolism in obesity are not yet understood. This study seeks to shed light on some of these mechanisms and it is part of a larger project.
The study and the data obtained are interesting. The authors should better outline the data obtained from their pilot study in such a way as to make some aspects of this manuscript clearer.

Author Response

Ans> Since the DNAJC6 gene was found by GWAS study to look for the significant genes between high- BMI with low-RMR and low-BMI with high-RMR, we assumed DNAJC6 could be related to obesity because of energy imbalance. However, we must start the primary CELL study in advance to prove the gene would be involved in obesity metabolism such as adipogenesis/lipolysis, inflammation, insulin resistance, and so on. Therefore, although the gene was found through human energy metabolism study and we measured O2 consumption to figure out energy consumption in mitochondria of DNAJC6-overexpressed cells, it is better to use “obesity metabolism” instead of “the energy metabolism” because of a part of obesity. 

Moreover, the outline of pilot study was published in Int J Pers Med (2021), which we did not mention well. However, we inserted this part in “Introduction”; line 63. 

Reviewer 2 Report

The study to show the role of DNAJC6 on adiposity is very novel and important. The work is designed and planned nicely. There are a few suggestions to improve the quality of the manuscript. The suggestions are given below:

a) The way line 114 is written about DAPI/BODIPY staining should be more explicit to get it clarified better. Currently it seems a continuation of ORO staining.

b) Line 162, the Cell Mito stress measuring equipment should carry the catalog number, brand details, and a specific name if any.

c) Figure 1A the RT-PCR gel picture is nicely showing the DNAJC6 overexpressing band but for 3T3-L1 no band is visible at all, should it be that way or one should expect a basal level expression, while that is the case in Western Blot data. Clarification needed.

d) In Figures 2, 3, and 4 there is an acronym MDI that I could not find anywhere explained in the text. Did I miss it?

e) In some places (e.g. line 291) the references are cited before the stop and in some cases after the stop. Please check the consistency and the journal requirement.

Reviewer 3 Report

Please find attached the document with my comments and suggestions.

Round 2

Reviewer 3 Report

Please find attached the comments.

Author Response

The authors reviewed and corrected several aspects of the manuscript, which has helped to improve it. However, there are several points that they did not consider or were not sufficiently clear.

  • pIRS and insulin signaling

Authors can conclude that when DNAJC6 is overexpressed, at day 8 there is a reduction in the amount of the insulin receptor and of IRS and that there is also a reduction in IRS1 inhibition (pIRS Ser307), thus IRS is more active. (1) With this idea they must improve the sections of Result, Discussion/Conclusion and Abstract. (2) At lines 331-332 this idea is incomplete. (3) At line 338 authors mentioned GLUT4 vehicle. What does thy mean with the word “Vehicle”. In case they refer to “vesicles”, this will not be appropriated since they only evaluated the total amount of GLUT4 protein content. Please clarify

ANS>

  • (1) In the results, according to the statics significance, we compared the actual results of IRS-1-AKT-GLUTS between Tg cells and the control cells. (line 253-261)

Since the data for distinct feedback pathways of ser307 phosphorylation of IRS-1 could not be performed by this experiment, it is better to focus on the comparison of Tg cells and control cells in normal conditions, and just introduced the possibilities of insulin resistance at ser307 phosphorylation in the conditions of prolonged exposure of insulin & dexamethasone, oxidative stress & so on. However, Potashink reported that factors-induced serine phosphorylation of IRS1 increases IRS-1 degradation (proteasome-independent pathway), but these changes cannot predict the induction of metabolic insulin resistance yet. [R Potashnik , A Bloch-Damti, N Bashan, A Rudich, IRS1 degradation and increased serine phosphorylation cannot predict the degree of metabolic insulin resistance induced by oxidative stress. Diabetologia, 2003 May; 46(5):639-48. DOI: 10.1007/s00125-003-1097-5]

In Tg cells, all protein expressions such as IR-beta, AKT, GLUT4 were depressed, and we concluded that insulin cascade signal to the glucose uptake was not working in Tg cells.

Without the experiments such as GLUT4 expression in vivo conditions of p-IRS Y896 treatment or fasting glucose, we did not mention the above theory. 

  • (2) Adipogenesis, triglyceride synthesis, in adipocytes requires the glucose influx through GLUT4 transporter on the membranes after insulin signal-cascade system was activated, such as IRS1-AKT-GLUT4 [22, 23]. At line 320-322.
  • (3) GLUT4 vehicle means GLUT4 transporter protein to influx glucose on membranes, which is recruited from intracellular stores when the IR-AKT-GLUT4 cascade system activates the IRS1 signals. We changed this text in discussion into “The reduction of insulin/receptor complex due to DNAJC6 overexpression subsequently suppressed the recruitments of the GLUT4 transporters from the intracellular stores.“ (At line 330-332)

  • Statistic

The way in which statistical information is being displayed is still deficient. Fig 1: the graphs show superscript letters corresponding to the ANOVA test, however in the figure legend it is also mentioned information about Student´s t-test that is nor represented in the graphs. In many graphs still appears “p=0.000” which is not adequate since statistic differences among groups are designated in the same graphs. The graphs that still have this information are: Fig 1B, 2A, 2B, 2C, 3A Lepr, 3B (insulin receptor), 4A, 4B and 4C.

ANS> T-test was used for RT-PCR in Fig 1A, but we excluded it because we did not show their graph. p=0.000 was changed by p<0.0001 in the ANOVA test, and the typo was corrected at line 206. (p<0.05). 

  • Adipokines

The term “secretion” for the adipokines must be substituted by “expression”, since they do not evaluate the adipokine concentration in the media. In most parts of the text it has been already substituted, however some are missing (line 30, 318).

ANS> In our labs, we usually detected adipokines in media with ELIZA kits; however, we did not check at this time. So “adipokines secretion:” was changed by “adipokines expression.”

  • LC3 II/I

At Line 159 and the last graph of Fig 4 still need to be corrected, since it still appears as LC3 I/II It should be even useful if authors can explain in a deeper manner the rationale of measuring LC3. It is not clear which graphs correspond to figure 4B and which ones to 4C. The figure needs more order

ANS>  We revised Fig 4C, 4B was for UCP1-expression, but 4C was for LC3-expression. Since the conversion of LC3-I into LC3-II was a significant biomarker for autophagy, we described it as the ratio of LC3-II and LC-I protein expression vs. b-actin in the Y-axis of Fig 4C graph. It was enormously increased by DNAJC6 overexpression compared to 3T3-L1 cells during adipocyte differentiation. We added them in the discussion part (at line 364-376).” Mizushima reported that simple comparison of LC3‑I and LC3‑II, or summation of LC3‑I and LC3‑II for ratio determinations, may not be appropriate, and instead, the amount of LC3‑II can be compared between samples. [37]” 

  • MDI

 I understand that MDI is used to refer to methylisobutylxanthine, dexamethasone and insulin that was used to induce adipocyte differentiation. And thus MDI- refers to day 0 and MDI+ refers to day 8, once cells are mature adipocytes. However, this information is not expressed openly and clearly, and not all readers will easily understand this technical detail. So, in the last revision I had requested to authors that in the figures, the terms MD1 -/+ should not be used, since in the text they always referred as day 0 and day 8. It is very important that in the figures, the differentiation days (0 or 8) appear below ALL the graphs instead of the signs (- or +). It is also important that the term MDI is clearly explained, since it is only mentioned in the methods (line 77) but is not sufficient.

ANS> We agreed that MDI(-) or MDI(+) were changed by day 0 or 8 for better understanding.